# Earthworm assemblages in different intensity of agricultural uses and their relation to edaphic variables

LB Falco[1], R Sandler[1], F Momo[1,2], C Di Ciocco[1], L Saravia[2] and C Coviella[1]

[1] Departamento de Ciencias Básicas, Programa de Investigaciones en Ecología Terrestre e INEDES, Universidad Nacional de Luján, Argentina
[2] Instituto de Ciencias, Universidad Nacional de General Sarmiento, Argentina

Corresponding author
LB Falco, falcoliliana@gmail.com

## ABSTRACT

The objective of this study was to relate earthworm assemblage structure with three different soil use intensities, and to indentify the physical, chemical, and microbiological soil variables that are associated to the observed differences. Three soil uses were evaluated: 1-Fifty year old naturalized grasslands, low use intensity; 2-Recent agricultural fields, intermediate use intensity, and 3-Fifty year old intensive agricultural fields, high use intensity. Three different sites for each soil use were evaluated from winter 2008 through summer 2011. Nine earthworm species were identified across all sampling sites. The sites shared five species: the native *Microscolex dubius*, and the introduced *Aporrectodea caliginosa*, *A. rosea*, *Octalasion cyaneum*, and *O. lacteum*, but they differed in relative abundance by soil use. The results show that the earthworm community structure is linked to and modulated by soil properties. Both species abundance and diversity showed significant differences depending on soil use intensity. A principal component analysis showed that species composition is closely related to the environmental variability. The ratio of native to exotic species was significantly lower in the intensive agricultural system when compared to the other two, lower disturbance systems. *Microscolex dubius* abundance was related to naturalized grasslands along with soil Ca, pH, mechanical resistance, and microbial respiration. *Aporrectodea caliginosa* abundance was related to high K levels, low enzymatic activity, slightly low pH, low Ca, and appeared related to the highly disturbed environment. *Eukerria stagnalis* and *Aporrectodea rosea*, commonly found in the recent agricultural system, were related to high soil moisture condition, low pH, low Ca and low enzymatic activity. These results show that earthworm assemblages can be good indicators of soil use intensities. In particular, *Microscolex dubius*, *Aporrectodea caliginosa*, and *Aporrectodea rosea*, showed different temporal patterns and species associations, due to the changes in soil properties attributable to soil use intensity, defined as the amount and type of agricultural operations.

## INTRODUCTION

The organisms living in the soil, collectively known as soil biota, play a crucial role in regulating processes like water infiltration and storage, decomposition and nutrient cycling, humus formation, nutrient transformation and transport; moreover, they stimulate the symbiotic activity in the soil, improve the organic matter storage, and prevent erosion (*Coleman & Crossley, 1996*; *Lavelle et al., 2006*).

Several of the ecosystem services provided by soil depend on the community of soil invertebrates (*Lavelle et al., 2006*), and earthworms are one of the most common components of edaphic communities. Earthworms are considered ecosystem engineers because they improve decomposition processes and nutrient cycling (*Lavelle et al., 1997*; *Six et al., 2004*) and have a strong effect on the soils' hydraulic properties (*Lee, 1985*; *Edwards & Bohlen, 1996*; *Lavelle & Spain, 2001*; *Lavelle et al., 2006*; *Johnson-Maynard, Umiker & Guy, 2007*; *Jouquet et al., 2008*). As key detritivores, earthworms are essential for soil nutrient recycling, and maintenance of soil structure (*Dennis et al., 2012*).

The most important factors limiting earthworm populations are food supply, moisture, temperature, and the physical and chemical characteristics of the soil such as pH, organic matter and macronutrients content (*Satchell, 1967*; *Lee, 1985*; *Curry, 2004*). Earthworm populations are also affected by the direct and indirect effects related to the type and extension of the vegetation cover (*Mather & Christensen, 1988*; *Falco & Momo, 1995*). Due to the strong relation between earthworms and soils (*Paoletti, 1998*), modern agricultural practices can modify the physical and chemical soil environment thus modulating changes in abundance and composition of earthworm communities (*Curry, Byrne & Schmidt, 2002*). In this regard, *Dale & Polasky (2007)* indicate that in agricultural systems, changes in land cover are the direct result of management practices. Moreover, the use of pesticides and herbicides in intensive agricultural systems is known to affect earthworms at different levels, from gene expression and physiology, to the individual, population, and community structure (*Pelosi et al., 2014*; *Santadino, Coviella & Momo, 2014*). In a study encompassing five different land use intensities in Colombia, *Feijoo et al. (2011)* found that high use intensity led to a loss of native species. Furthermore, the use of heavy machinery prevents these native species from re-colonizing due to soil compaction. Therefore, when changes occur in agricultural practices, earthworm assemblages are able to respond to the ensuing changes in the soil's physical properties and environmental conditions (*Lavelle et al., 1997*; *Johnson-Maynard, Umiker & Guy, 2007*).

Since earthworm abundance and distribution are strongly influenced by the environmental conditions and the ecological status of the system (*Paoletti, 1998*; *Falco & Momo, 2010*; *Pelosi et al., 2015*), the earthworm community structure can be successfully used as a biological indicator of soil conditions (*Paoletti, 1999*; *Momo, Falco & Craig, 2003*). *Guéi & Tondoh (2012)* found that earthworms can be used to monitor land-use types with different levels of soil quality.

The use of bioindicators has the advantage of providing historical and functional information about soils. The earthworm community structure integrates this information on soil conditions both in space and time and provides indications of the soil's ecological state.

In this context, the objectives of this study were: (1) To assess the earthworm community structure under three different soil use intensities: Fifty years old intensive agriculture, two years old recent agriculture, and naturalized grasslands. (2) To identify the physical, chemical, and microbiological variables related to the different soil use intensities affecting the earthworm community structure. (3) To detect which earthworm species are typical of each set of soil conditions as affected by use intensity.

## MATERIALS AND METHODS

### Sampling sites

This study was performed in the rolling pampas within the Argentine pampas, a wide plain with more than 52 million hectares (520.000 km$^2$) of land suitable for cattle production and cropping (*Viglizzo et al., 2004*). It is one of the largest and most productive agricultural regions in the world.

Agricultural systems based on crop–crop and crop–pasture rotations under grazing conditions have been very common in the region for over a century until the 1980s. The adoption of conservative tillage and no-till practices has significantly increased during the 1980s and 1990s. Although pesticides were extensively used since the 1960s, crops and pasture fertilization increased noticeably only during the 1990s (*Viglizzo et al., 2003*). The expansion of the land area used for annual crops means that the pampean ecosystem is currently under an intense disturbance regime (*Navarrete et al., 2009*).

The selected study sites are located in central Argentina, on Argiudolls soil (Mollisols, Typic Argiudolls (*Soil Survey Staff, 2010*)). The study sites were privately owned fields located in Navarro, Buenos Aires Province (34°49′35″S, 59°10°38″ W), and Chivilcoy (35°03′10″ S; 59°41′08″ W) approximately 75 and 150 km west of Buenos Aires City, respectively (Fig. 1).

The weather regime in this region is temperate humid, with a mean annual rainfall around 1,000 mm. The mean annual temperature is 17 °C. Phytogeographycally, it is within the neotropical region, oriental district of the Pampean Province, and the dominant vegetation is a gramineous steppe (*Cabrera & Willink, 1973*).

### Land use intensity in the selected sites

The analyzed systems differed only in their use intensity, as defined by the amount and type of agricultural operations, such as tillage, pesticide use, rotation, fertilization, and harvesting. Samplings were carried out on three different types of soil uses (agro-ecosystems) which represent three different levels of disturbance of the same Argiudoll. These are:

High disturbance agro-ecosystem: Intensive agricultural system (AG); sites with 50 years of continuous intensive agricultural practices, under a regular corn-wheat-soybean rotation, currently under no-tillage, and chemical weed controls are used. During cropping season, heavy machinery is used and insecticides, herbicides, and fertilizers are applied several times a year.

Intermediate disturbance agro-ecosystem: recent agricultural system (RA); cattle-grazing sites that were under direct grazing for 40 years. Originally managed under direct

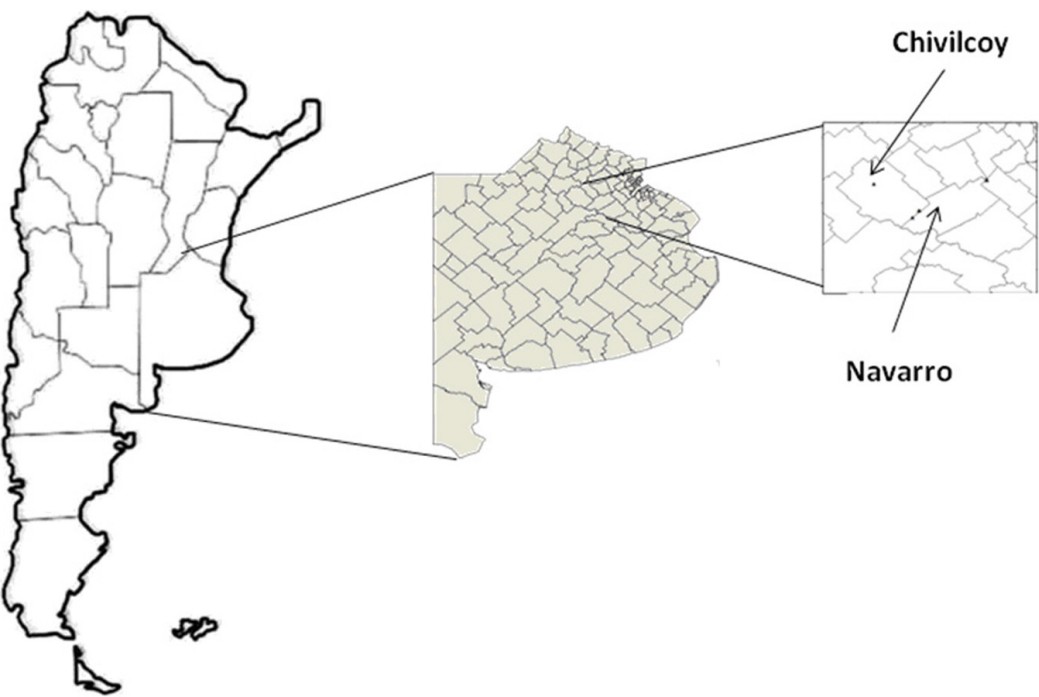

**Figure 1 Sampling sites.** Map showing the location of the sampling sites.

grazing, these sites moved to bale production (oat, maize, and sorghum) two years before the start of this study.

Low disturbance agro-ecosystem: Naturalized grasslands (NG); sites with no significant anthropic impact during the last 50 years.

Nine observation fields (three replicates for each one of the three agricultural systems) were evaluated. Replicates were separated from each other by at least several hundred meters to a few kilometers. At each site, five samples were taken every three months for a period of two years.

At each sampling date, five random samples were taken 25 m apart from each other per each replicate (3) and treatment (3). Thus, a total of 45 samples were taken per sampling date. The size of each sample was of $25 \times 25 \times 25$ cm.

The measured environmental variables were bulk density (BD), mechanical resistance (MR), gravimetric moisture content ($w$), electrical conductivity (EC), organic mater (OM), pH, total N, total P, exchangeable Ca, exchangeable Mg, exchangeable K, and exchangeable Na. To characterize the sites, microbiological activity was assessed through soil respiration and free nitrogen-fixing bacteria activity (Nitrogenase Acetylene Reduction Activity, ARA). Methods used for chemical and physical analyses are shown in Table 1.

Earthworm extraction from the soil samples was performed by handsorting. Earthworms were preserved with soil until identification in the laboratory. Later they were fixed and preserved in alcohol–formalin–glycerin, following *Righi (1979)*, and identified by external morphology using keys from *Righi (1979)* and Reynolds (*Reynolds, 1996*). Clitelated individuals were identified to species level and pre-clitelated ones to genus. At

**Table 1 Physicochemical and microbiological parameters measured ($n = 150$ per system).** Different letters within a row indicate significant differences between systems, $P < 0.05$ Kruskall–Wallis ANOVA tests. Means ± SE shown.

| Parameters | Method | System | | |
|---|---|---|---|---|
| | | NG | RA | AG |
| Available P (ppm) | Kurtz and Bray | 11 ± 8.5 b | 15 ± 12 a | 14 ± 12 a |
| OM (%) | Walkey–Black | 4 ± 1.5 a | 4 ± 1.5 a | 4 ± 1.4 a |
| EC (dS m$^{-1}$) | Conductivimeter | 1,5 ± 1.3 a | 0.8 ± 0.5 b | 0.7 ± 0.5 c |
| pH | 1:2 soil to water ratio | 7.5 ± 1.0 a | 6 ± 0.6 b | 6 ± 0.5 b |
| Bulk density (g cm$^{-3}$) | Cylinder method | 1.2 ± 0.2 a | 1.1 ± 0,1 b | 1.2 ± 0.1 a |
| Moisture content (g water g soil$^{-1}$) | Gravimetric | 0.2 ± 0.1 a | 0.3 ± 0.1 b | 0.2 ± 0.1 a |
| Exch. Ca (cmol kg soil$^{-1}$) | Titration with EDTA | 6.7 ± 1.3 a | 5 ± 0.5 b | 6 ± 0.7 a |
| Exch. Mg (cmol kg soil$^{-1}$) | Titration with EDTA | 1.8 ± 0.4 a | 1.5 ± 0.7 b | 1.6 ± 0.5 b |
| Exch. Na (cmol kg soil$^{-1}$) | Flame photometry | 1.3 ± 0.5 a | 0.8 ± 0.2 b | 0.7 ± 0.2 c |
| Exch. K (cmol kg soil$^{-1}$) | Flame photometry | 1.6 ± 0.5 a | 1.3 ± 0.3 b | 1.6 ± 0.5 a |
| Total N (%) | Kjeldahl | 0.28 ± 0.1 a | 0.32 ± 0.1 b | 0.29 ± 0.05 b |
| Nitrogenase activity (nanolitres of ethylene g dry soil incubation hour $^{-1}$) | ARA | 0.3 ± 0.3 a | 0.2 ± 0.2 b | 0.2 ± 0.3 b |
| Respiration (mg CO$_2$ g dry soil day$^{-1}$) | Alkaline incubation | 0.09 ± 0.06 a | 0.07 ± 0.05 b | 0.05 ± 0.05 c |
| MR 0–5 cm (kg cm$^{-2}$) | Cone penetrometer | 10 ± 6 a | 2.5 ± 3 b | 5.5 ± 4 c |
| MR 5-10 cm (kg cm$^{-2}$) | Cone penetrometer | 13 ± 7 a | 5 ± 5 b | 8 ± 5 c |

**Notes.**

NG, Naturalized grassland; RA, Recent agriculture; AG, Intensive agricultural system; P, Phosphorus; OM, Organic matter; EC, Electrical conductivity; Ca, Calcium; Mg, Magnesium; Na, Sodium; K, Potassium; MR, Mechanical resistance; ARA, Acetylene Reduction Activity.

each site, earthworm taxonomic composition and population density were measured. Earthworm communities were characterized at each soil use intensity by population density, species richness both observed and estimated using the Chao index (*Magurran, 2004*), and diversity using the Shannon index (*Zar, 1999*).

The Chao index was calculated as:

$$\hat{S} = S_{\text{obs}} + (a^2/2b)$$

where:

$\hat{S}$: Species richness estimate

$S_{\text{obs}}$: Observed species richness

$a$: Number of species found in only one sample, and

$b$: Number of species found in only two samples.

The Shannon index ($H'$) was calculated as:

$$H' : -\sum p_i * \log_2 p_i$$

where:

$p_i$: Number of individuals belonging to the species $i$/total number of individuals.

## Statistical analyses

Due to the non-normal distributions of the physical and chemical data, Kruskall–Wallis ANOVA tests were carried out to compare variables between treatments. The Shannon index comparisons were performed using ANOVA.

The relationship between environmental variables and earthworm species abundances was analyzed by means of a principal component analysis (PCA) using abundances. Prior to analysis, the species abundances data were transformed using the Hellinger distances (*Legendre & Gallagher, 2001*) in order to preserve the distances among samples. To calculate the Hellinger distances, the abundances are first divided by the sample total, the result is square root transformed, then Euclidian distances are computed. As a result, Hellinger distances are scalar measurements of the divergence  in the distribution of samples. These distances are then used in the PCA. Physical and chemical variables were then fitted into the ordination space described by the first two principal components of the earthworm data by projecting biplot vectors. The statistical significance of the environmental variables is based on random permutations of the data and P-values were adjusted by a sequential multiple test procedure of *Hommel (1988)*. The ordination analysis and vector fitting were produced using the R statistical language (*R Core Team, 2012*) and the Vegan package (*Oksanen et al., 2011*).

The relationship between the characteristics of the environment and earthworm presence was further analyzed at the genus level, assessing the sensitivity of the groups with the soil parameter values through a Mann–Whitney U-test. The program Statistica 7.0 (Stat Soft, Inc., Tulsa, Oklahoma, USA) was used.

## RESULTS

### Physical and chemical soil parameters

Of all the physical–chemical and microbiological parameters evaluated, only four variables: exchangeable Sodium (Na), Electrical conductivity (EC), Mechanical resistance (MR), and respiration showed significant statistical differences between each of the three systems and only organic matter (OM) presented no differences (Table 1). From the four variables that separate the three systems, the naturalized grasslands showed the highest Na levels and EC values.

Microbiological activity and soil microfauna were assessed through soil respiration and nitrogen fixing bacteria activity, which separated the naturalized grasslands for their high value when compared to the other two agroecosystems. Naturalized grasslands were also separated from the other two agroecosystems by having a much higher microbiological and soil microfauna activity.

### Earthworm assemblage response to soil use intensity

Results show that each soil use presents a different species composition and abundance (Fig. 2). The relative abundances of the earthworm species found in each system are shown in Fig. 3.

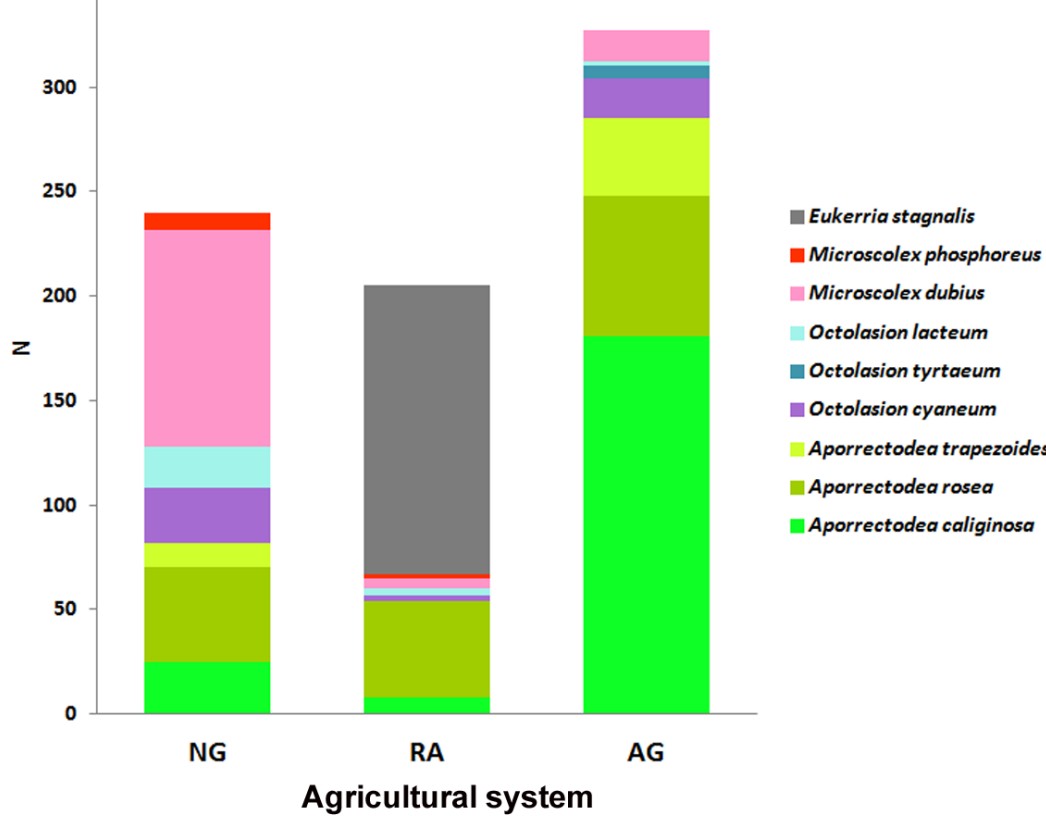

**Figure 2 Earthworm species per system.** Abundance (N) of each earthworm species throughout the total sampling period for each system. NG, Naturalized grasslands; RA, Recent agriculture; AG, Intensive agriculture.

**Table 2 Ecological parameters measured.** Observed and estimated species richness, mean density, and Shannon diversity index. $S_{obs}$, Observed richness. Different letters in the Shannon index column indicate significant differences (one-way ANOVA $p < 0.05$).

|  | Richness observed ($S_{obs}$) | Richness estimate (Chao) | Density (individuals/m$^2$) | Shannon index |
|---|---|---|---|---|
| Naturalized grassland | 7 | $7 \pm 0$ | $46 \pm 19$ | 0.53 a |
| Recent agriculture | 7 | $8.5 \pm 1.5$ | $40 \pm 55$ | 0.37 b |
| Intensive agricultural system | 7 | $7.25 \pm 0.4$ | $76 \pm 56$ | 0.57 a |

A total of 9 earthworm species were identified across all systems. Five species were common to all of them: the native *Microscolex dubius* (Fletcher, 1887) and the exotic *Aporrectodea caliginosa* (Savigny, 1826), *A. rosea* (Savigny, 1826), *Octolasion cyaneum* (Savigny, 1826), and *O. lacteum* (Oerley, 1885), but they differed in their abundances. The differences in abundance explain the significant differences found for the Shannon index values (ANOVA test $p < 0.05$). The richness estimate ($\hat{S}$) and the observed richness ($S_{obs}$) only differed in the recent agricultural system (Table 2).

In the naturalized grasslands, the species identified as being the dominant (44% of all the individuals collected) was the epigeic native *M. dubius*, followed by the endogeic

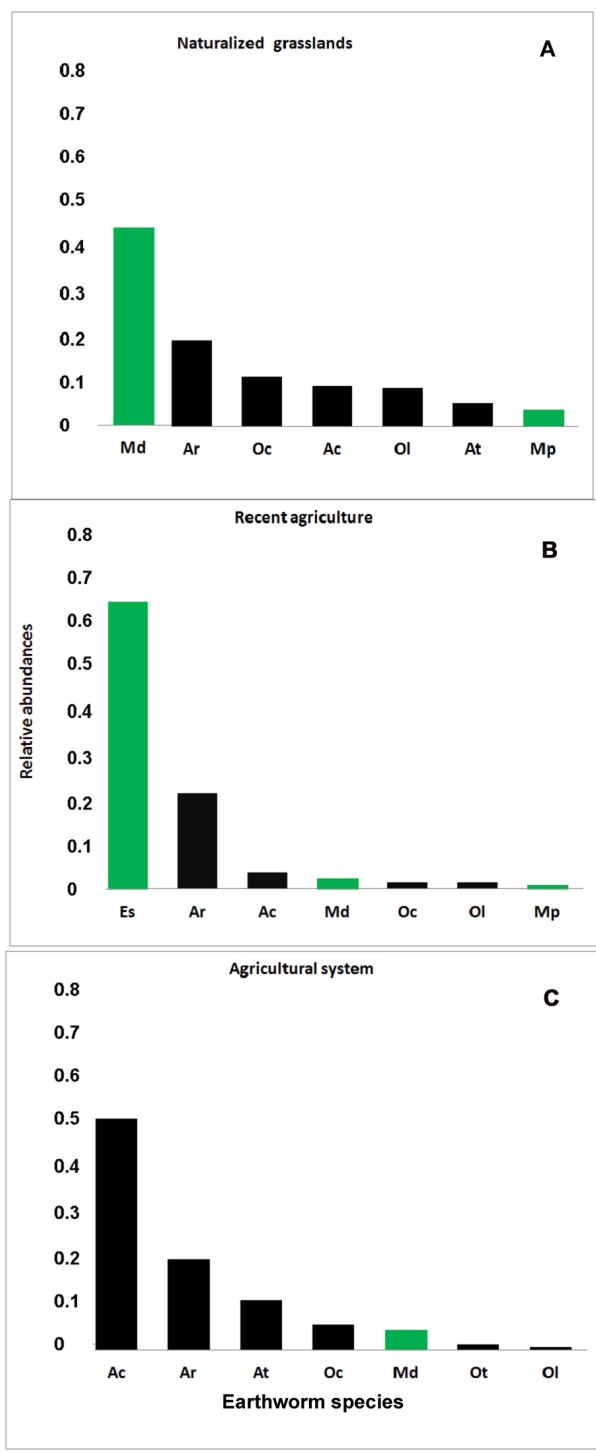

**Figure 3 Rank-abundance plot.** Relative abundance (ni/N) for each earthworm species identified in the three systems. (A) Naturalized grassland (NG), (B) Recent agricultural (RA), (C) Agricultural system (AG). Ac, *A. caliginosa*; Ar, *A. rosea*; At, *A. trapezoides*; Ot, *O. tyrtaeum*; Md, *M. dubius*; Oc, *O. cyaneum*; Ol, *O. lactuem*; Es, *E. stagnalis*; Mp, *M. phosphoreus*. Native species shown in green, exotics in black.

exotics *A. caliginosa*, *A. rosea*, *O. cyaneum*, and *O. lacteum*. The other endogeic exotic species, *Aporrectodea trapezoides* (Dugés, 1828), and the native *M. phosphoreus* (Dugés, 1837), were less frequent (Fig. 3A). Forty seven percent of all the individuals collected belonged to native species, and the ratio natives/exotics was 2:5.

In the recent agricultural system, the endogeic native *Eukerria stagnalis* (Kinberg, 1867) was dominant and the exotic *A. rosea* was also common. Other species that were present, albeit with a low frequency, were *A. caliginosa,* and *M. dubius. M. phosphoreus, O. cyaneum*, and *O. lacteum* appeared on either one or two sampling dates only. In this system, *E. stagnalis* represents 68% of all the individuals collected and *A. rosea* represents 22% (Fig. 3B). The ratio of native-exotic species was 3:4.

In the intensive agricultural system, the most common species were the endogeic exotics *A. caliginosa*, *A. rosea*, and *A. trapezoides*. The other endogeic species *O. cyaneum*, and the epigeic native *M. dubius* were less frequent. *Octolasion tyrtaeum* (Savigny, 1826) was only detected in the first sampling date, and *O. lacteum* appeared in two sampling dates with a single individual each date. Here, the exotic species represent 95% of the individuals (Fig. 3C), with *M. dubius* being the only native present in this system. The agricultural system had the lowest ratio of native-exotic species (1:6).

The differences in the chemical and physical soil parameters and the species requirements determined the species' co-occurrences found in each system. We observed these associations involving both native and introduced species, and combining different ecological categories. The associations most frequently found in naturalized grasslands were: *A. rosea—M. dubius* (appearing together in 33% of the samples), *O. cyaneum—O. lacteum* (10%), and *A. rosea—O. cyaneum* (10%). In the recent agricultural sites, *A. rosea—E. stagnalis* were found together in 67% of the samples, and in the intensive agricultural system, the most common associations were *A. caliginosa—A. rosea* (12.5%), and *A. rosea—M. dubius* (12.5%).

The relationship between the characteristics of the environment and earthworm presence, assessing the sensitivity of the groups with the soil parameter values is shown in Table 3.

*Aporrectodea*, *Octolasion*, and *Microscolex* genus were present in samples with the same levels of Mg, K, and BD. *Octolasion* separated from *Aporrectodea* only because of Ca levels, and its response to soil moisture, MR, and respiration put it close to *Microscolex*. In turn, *Microscolex* differed from the other groups due to Na, pH, ARA, and high MR (MR 10 cm). On the other hand, *Eukerria* was related to places with low levels of Ca, K, pH, EC, ARA, BD, MR and moisture content.

In order to know how the species' composition explains the environmental variability, an indirect ordination PCA analysis was used, followed by a vector fitting (Fig. 4). Interestingly, the analysis showed no relationship between species and fertility levels (N, P, and OM), but it did show a relationship with exchangeable Mg and Ca.

The first two axes explain 58% of the variance. The environmental variables that were significantly related to the species ordination were: moisture, K, ARA, respiration, MR, Ca, and pH (adjusted $P < 0.05$).

**Table 3 Variables measured at the sampling points where each earthworm genus was recorded.** Mean (range) values of each measured variable as they relate to earthworm genus presence (Non-clitelated specimens included). Different letters within each row indicate significant differences between earthworm genus, $P < 0.05$ Mann–Whitney U-test pairwise comparisons. Abbreviations as in Table 1.

| Parameter | Aporrectodea | Octolasion | Microscolex | Eukerria |
|---|---|---|---|---|
| OM (%) | 4.4 (3.7–5.3) a | 4.8 (4.4–6.1) a | 4.9 (3.2–5.9) a | 4.7 (4–5.9) a |
| N (%) | 0.29 (0.26–0.33) a | 0.29 (0.26–.33) a | 0.29 (0.27–0.34) a | 0.29 (0.25–0.34) a |
| Available P (ppm) | 8.7 (4.4–17.6) a | 7.7 (3.6–15.2) a | 9.3 (4.8–15) a | 6.8 (4.4–14.1) a |
| Exch. Ca (cmol kg soil$^{-1}$) | 6.0 (5.5–6.4) a | 6.6 (6.1–9) b | 6.1 (5.8–7) c | 5 (4.6–5.4) d |
| Exch. Mg (cmol kg soil$^{-1}$) | 1.7 (1.1–2) a | 1.7 (1.1–1.9) a | 1.6 (1.5–1.9) a | 1.1 (1–1.6) b |
| Exch. Na (cmol kg soil$^{-1}$) | 0.8 (0.7–1) a | 0.74 (0.4–0.9) a | 0.9 (0.7–1.1) b | 0.8 (0.7–1.1) a |
| Exch. K (cmol kg soil$^{-1}$) | 1.3 (1.1–1.7) a | 1.5 (1.2–1.8) a | 1.3 (1.1–1.8) a | 1.1 (1–1.4) b |
| pH | 6.2 (5.8–7) a | 6.3 (6–6.8) a | 6.8 (6.2–7.2) b | 6 (5.6–6.5) c |
| EC (dS m$^{-1}$) | 0.6 (0.3–0.9) a | 0.6 (0.3–0.9) a | 0.7 (0.3–1.2) a | 0.4 (0.2–0.7) b |
| Nitrogenase activity (nanolitres of ethylene g dry soil incubation hour $^{-1}$) | 0.15 (0.07–0.3) a | 0.26 (0.11–0.35) a | 0.27 (0.14–0.37) b | 0.15 (0.12–0.18) a |
| Respiration (mg CO$_2$ g dry soil day$^{-1}$) | 0.04 (0.03–0.09) a | 0.04 (0.03–0.09) ab | 0.07 (0.04–0.1) b | 0.05 (0.02–0.07) a |
| Moisture content (%) | 0.26 (0.2–0.3) a | 0.23 (0.17–0.29) ab | 0.24 (0.2–0.3) b | 0.33 (0.3–0.4) c |
| Bulk density (g cm$^{-3}$) | 1.2 (1.1–1.3) a | 1.21 (1.1–1.3) a | 1.2 (1.1–1.3) a | 1.1 (1–1.2) b |
| MR 0-5 (kg/cm$^2$) | 4.6 (2.25–8.2) a | 4.9 (3–8) ab | 7.8 (4.3–12.5) b | 0.78 (0–3) c |
| MR 5 = 10 (kg/cm$^2$) | 6.5 (3.5–10.8) a | 7.6 (4–11.5) a | 10 (7–17) b | 2.6 (0.8–5.5) c |

As shown in Fig. 4, the ordination method shows that *M. dubius* densities appeared related to the levels of Ca, pH, MR and respiration. This species is well adapted to environments rich in Ca, neutral pH, high microbiological activity, and high mechanical resistance. The environment defined by *M. dubius* was related to the characteristics of the Naturalized grassland system, and this species can be considered as indicative of the conditions dominant in this system. In the same way, *A. caliginosa* (Fig. 4) is related to high K levels, low enzymatic activity, low pH, and low Ca. Such soil parameters are characteristic of the Intensive agricultural system, thus making *A. caliginosa*, a natural cosmopolite and invasive species, a good indicator of high perturbation sites. Finally, *E. stagnalis* and *A. rosea* were related to the second ordination factor, and they describe an environment with high soil moisture, low pH, low Ca levels, and low ARA. These characteristics describe the Recent agricultural system.

## DISCUSSION

These results show that the structure of earthworm assemblages change in relation to differences in soil use intensity in terms of its composition, abundance, and species associations. The data presented here shows that, in the same soil and the same regimen of temperature and precipitation, the earthworm assemblage composition and abundance varied across the different systems studied. These variations describe the physical and chemical differences of soil due to land use intensities and their associated management practices (*Geissen, Peña-Peña & Huerta, 2009*).

Tillage, weed control, fertilization and soil cover are parameters that best characterize the different land use intensities (*Curry, 2004*; *Viglizzo et al., 2004*; *Decaëns et al., 2008*),

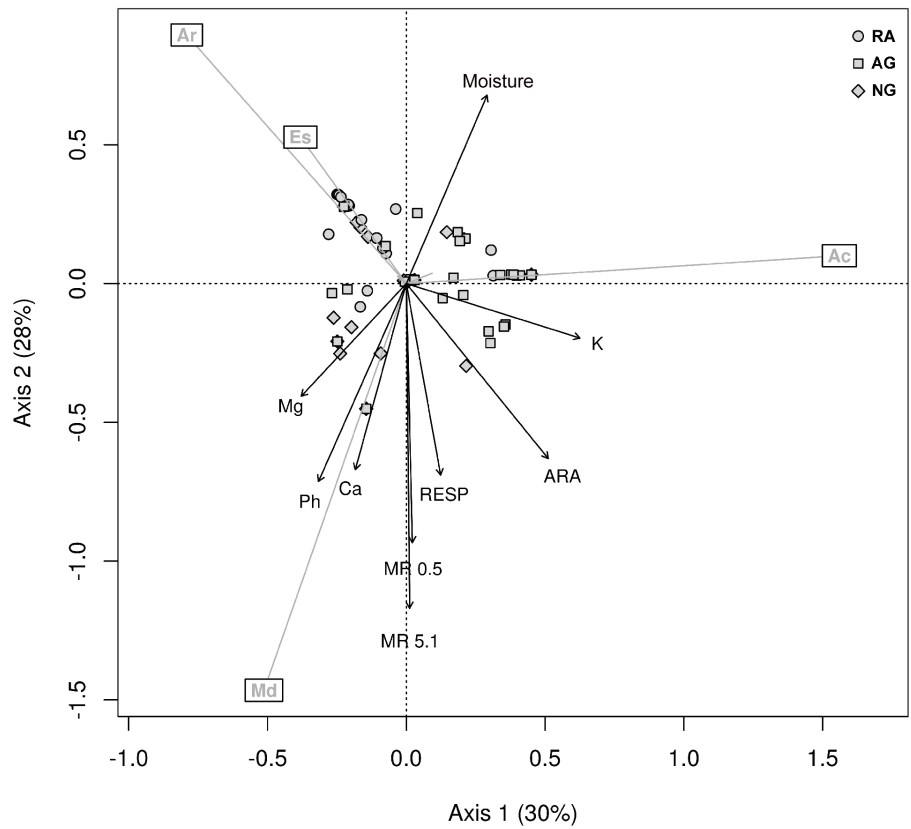

**Figure 4 Principal component analysis.** PCA biplot of Hellinger transformed earthworm species, only the four most abundant ones are shown. The arrows are significant environmental variables fitted into the ordination space. The percentage of explained variance is shown in each axis. Ac, *A. caliginosa*; Ar, *A. rosea*; Es, *E. stagnalis*; Md, *M. dubius*.

modifying the physical (water and air movement) and chemical environment, thus changing the habitat's suitability.

In the agricultural system under highest use intensity, earthworm communities were affected directly by the changes caused by tillage practices or indirectly through changes in food supply. Several studies indicate that earthworm communities are more abundant and rich in species in undisturbed soils when compared to cropland (*Emmerling, 2001*; *Curry et al., 2008*; *Decaëns et al., 2008*; *Feijoo et al., 2011*; *Felten & Emmerling, 2011*). In this study, however, this pattern was not observed. All three systems have the same richness value but the abundances are consistently higher in the AG system with the highest use intensity. This system also showed the highest native species substitution by exotic ones (ratio 1:1.6). Total individuals belonging to invasive species varied from 16.6% in the NG system to 40.4% in the AG system. These results agree with those of *Lee (1985)*, *Paoletti (1999)*, and *Smith et al. (2008)*, who found that annual croplands have higher earthworm abundance than older fields. The dominance of introduced species is another characteristic of highly disturbed sites, as pointed out by *Fragoso et al. (1999)*, *Winsome et al. (2006)*, and *Chan & Barchia (2007)*.

The results presented in this work indicate that the response of the earthworm assemblage to the same soil subjected to different use intensities can be used as an agroecosystem biological indicator. This response can be explained in terms of quality and quantity of food (*Bohlen et al., 1997*), the long term use of inorganic fertilizers which have a positive effect on the total number of worms (*Edwards & Bohlen, 1996*; *Curry, 2004*), pH changes, and the level of Ca in the soil (*Lee, 1985*; *Paoletti, 1999*; *Smith et al., 2008*).

In the intensive agricultural and recent agricultural systems, microbiological activity was low when compared to the naturalized grasslands. This can be explained as the result of a reduction in pH and Ca, as well as by the ecological categories of the earthworm species present (*Scheu et al., 2002*). Indeed, *Scheu et al. (2003)* indicated that the presence of endogean species significantly reduces bacterial biomass and the functioning of the microbial assemblage. In the AG system, 95% of the species present are exotic endogeans, while in the RA system 97% are endogean (70% native, 30% exotic).

Soil use intensity was also indicated by the presence of a few species that were closely related to environmental variability. The intensification of the agricultural activities in the Pampas determined up to a 50% reduction in the calcium level (*Casas, 2005*). The ordination analysis related *M. dubius* with high Ca levels, and thus to less disturbed environments.

In fact, *Mele & Carter (1999)* point out that the distribution and number of native species are negatively correlated with P, K, and Mg levels, since these species are adapted to lower nutrient levels. In our study, the only species that is related to higher K levels is *A. caliginosa*, which is the most abundant earthworm in the intensive agricultural system.

## CONCLUSIONS

The data from this study indicate that the three agricultural systems are different in terms of the levels of exchangeable cations (Ca, K), pH, microbiological activity, and physical variables such as mechanical resistance and moisture. While the Argiudoll soil type is the same for all three systems, changes in land use intensity caused the observed differences in the soil's chemical and physical properties. Earthworm species assemblages also reflected the changes in these variables and are therefore good indicators of the systems studied.

The high diversity and highest number of earthworms found in the agricultural system under no tillage are consistent with the results by *Pelosi, Bertrand & Roger-Estrade (2009)*, who found that soil tillage and surface mulch are important parameters for the study of the effects of agricultural practices on earthworm communities.

*Microscolex dubius* was associated with sites exhibiting high levels of Ca, microbiological activity, and high mechanical resistance; thus describing the naturalized grassland system. *Eukerria stagnalis* is primarily associated with high moisture condition, and *A. caliginosa* is associated with highly disturbed environments, those with high K levels, low EC and Na, and low microbiological activity, all typical of the intensive agricultural system.

*Eukerria stagnalis* is indicative of high moisture condition, increased soil acidity, and a reduction in the levels of calcium and potassium, which are conditions prevalent in the intermediate use intensity system.

*Aporrectodea caliginosa* is the species best adapted to the most disturbed environment. This implies that the population recovers quickly after a disturbance (*Curry, 2004*; *Felten & Emmerling, 2011*; *Decaëns et al., 2011*), as it is known not to be significantly affected by changes in litter quality (*Curry & Schmidt, 2007*).

It is interesting to note that the earthworm species most related to the different systems are not linked to the variables most usually measured: OM, N, and P. Therefore, monitoring these species would provide indirect indication of nutrient variables, such as Mg, Ca or K (Fig. 4), thus complementing the information provided by other more common soil analyses in agro-ecosystems.

The patterns in the distribution and abundance of earthworm species observed in this study followed the differences in the physical and chemical variables measured in the different systems. An important difference between the studied systems is related to the agrochemicals used in the AG system. Earthworms are strongly affected by herbicide and insecticide use (*Pelosi et al., 2014*; *Santadino, Coviella & Momo, 2014*), and was likely also reflected in the differences in earthworm composition observed in the present study. All of these differences are, in turn, a reflection of the different management practices applied to the same Argiudol.

The richness, composition and abundance, as well as the species associations found, reflected the physical, chemical, and biological changes brought about as a result of the different intensities of the agricultural practices used in each system. The high abundance of the native *M. dubius* was associated with less anthropic activity. As a result, a strong population density reduction of this species can be interpreted as indicative of a high disturbance regime. On the other hand, *A. caliginosa* increased its density as disturbance increased. The presence of *A. caliginosa,* is clearly associated with highly disturbed environments as it was indeed found in several other published works (*Johnston et al., 2014*; *Lüscher et al., 2014*).

An increase in soil use intensity leads to changes in the physical and chemical properties of the same original soil. Earthworm assemblages are then affected by two main mechanisms. Firstly, by the agrochemical and mechanical perturbations introduced by intensive agricultural practices. Secondly, by the effects that the changes in the chemical variables have on earthworm assemblages (e.g., Ca content). On the other hand, as ecosystem engineers, earthworms affect edaphic variables, for example, microbiological activity. Therefore, the association between presence and abundance of the different earthworm species can be used as a biological indicator of the physical and chemical conditions of the soil they inhabit.

These results show that the structure of the earthworm assemblages can be reliably used for monitoring different soil use intensities.

## ACKNOWLEDGEMENTS

The authors wish to acknowledge the collaboration of Edgardo Ferrari, Pablo Peretto, and Romina de Luca for allowing the use of their fields as sampling sites. Ms. Loreta Gimenez greatly helped with field and laboratory works. The help of Dr. John T. Eigenbrode and

Dr. Beth Frankel with previous English versions, and that of Dr. Mark Breidenbaugh with the final revision of the English version of this manuscript are greatly appreciated. The revision of previous drafts by Dr. Adonis Giorgi is also appreciated.

### Funding

This work was partially funded by a grant from the Ministry of Science and Technology of Argentina, Project PICT-02293-2006, grant UBACyT number G074, and by the Universidad Nacional de Luján. The funders had no role in study design, data collection and analysis, decision to publish, or preparation of the manuscript.

### Grant Disclosures

The following grant information was disclosed by the authors:
Ministry of Science and Technology of Argentina: PICT-02293-2006.
UBACyT: G074.
Universidad Nacional de Luján.

### Competing Interests

The authors declare there are no competing interests.

### Author Contributions

- LB Falco and R Sandler conceived and designed the experiments, performed the experiments, analyzed the data, wrote the paper, prepared figures and/or tables, reviewed drafts of the paper, field work.
- F Momo analyzed the data, wrote the paper, prepared figures and/or tables, reviewed drafts of the paper.
- C Di Ciocco analyzed the data, wrote the paper, reviewed drafts of the paper, field work.
- L Saravia analyzed the data, wrote the paper, reviewed drafts of the paper.
- C Coviella conceived and designed the experiments, performed the experiments, analyzed the data, wrote the paper, reviewed drafts of the paper, field work.

### Supplemental Information

Supplemental information for this article can be found online at http://dx.doi.org/10.7717/peerj.979#supplemental-information.

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
