# Peer review of "Earthworm assemblages in different intensity of agricultural uses and their relation to edaphic variables"

_PeerJ, doi:10.7717/peerj.979_

## Round 0.1 · original submission · Major Revisions

The reviewers both agree that this is an interesting study. However the text needs to be proof read by an English native speaker. Please use period "." for a decimal mark (Tables 1- 3). Table 1 needs a revision, many expression is still in Spanish , such as "Porta" "y".

Specific comments:
- Line 134 What is Chao? and Shannon? Please define
- Line 143. What is Hellinger transformation
- Line 155-156. Please specify clearly what is Na, EC, MR
It was concluded that earthworm assemblage changes in relation to
differences in soil use. However it was also found that certain soil properties control the distribution of the species. How can the effect of soil properties and land use be separated?

Reviewer 1 ·

Basic reporting

Overall assessment

The paper presents an analysis on the relation between farming system practices/soil use and the assemblage of earthworm population. Due to the importance of preserving soil health, the topic is surely very important. This work then may offer some new information. However, in order to be considered for publication it requires a major revision of the present text, and some more and important details need to be supplied as well.

Some issues that should be dealt with by the authors:

* Text and references need careful editing

* Background information to demonstrate how the work fits into and add up to the field, and recent relevant literature are missing

* Cattle-grazing: I have problems with this system, what you sampled is not a cattle-grazing system but two yeas old cereal fields, where before there was cattle-grazing, that is not the same thing. This should be clearly pointed out. That also in view of your statement that populations adapt rapidly to new conditions.

* Effect of pesticides/herbicides? It seems that this is a very important issue
Pesticides and earthworms. A review
Céline Pelosi et al. Agron. Sustain. Dev. DOI 10.1007/s13593-013-0151-z

* It would be interesting also to discuss the effect and role of invasive species, e.g. if more species means more invasive species is that good? What's the role of invasive species in relation to the native (are the latter at risk?)




Details

Text needs some editing, some typos and mistakes

e.g. Abstract
3- Fifty year = years
disturbance Systems = systems
tos oil use = top soil use
found un the cattle-grazing = in

e.g. text
(Table 1),.



Key words
I'd add some more key words such as, earthworms, bioindicator, farming systems
I'd delete soil biota, as the paper is specifically dealing with earthworms

Scientific names, at least on the first appearance, should be completed with the name of the author who described the species



In the test references are reported in many different ways, a unique style should be used

(Curry, Byrne and Schmidt, 2002)
“and” should be &

(Satchell, 1967; Lee, 1985; Curry, 2004). older to newer
(Lee, 1985; Edwards and Bohlen, 1996; Lavelle
and Spain, 2001; Lavelle et al., 2006; Johnson-Maynard, Umiker, and Guy, 2007; Jouquet et al,
2008). random order


References need editing, many mistakes

Winsome T, L Epstein, PF Hendrix, WR Horwath,(2006) Competitive interactions between native
and exotic earthworm species as influenced by habitat quality in a California grassland.
Agric., Ecosyst. Environ., Appl. Soil Ecol. 32 :38–53.

two journals!? (mistake repeated many times)


Oikos. 101:225-238.
Agric Ecosyst Environ.101, 39–51.
Appl. Soil Ecol. 32 :38–53.

three different styles

Please use the same stile for books' page

Some papers use different fonts


Table 1
a,b,c, = ?
Respiration (mg de CO2*gr dry soil day-1) = “de” is Castellano I think


Table 2
Cattle-grazing: I have problems with this system, what you sampled is not a cattle-grazing system but tow yeas old cereal fields, where before there was cattle-grazing, that is not the same thing. This should be clearly pointed out. That also in view of your statement that populations adapt rapidly to new conditions.

It would be interesting to report also the ratio invasive/ indigenous (number and number of species) and discuss whether the there are any effects due to the invasive species (e.g. loss of native species, or reduction of their population)


Table 3
a,b,c = ?

Respiration (mg de CO2*gr dry soil day-1) = “de” is Castellano I think


Fig 1
what's in y axis?
NG, CG, AG, legend, it can facilitate a quicker reading



TEXT

46 Soil biota many be clearly defined to the non specialist readers


70-71 on EW as bioindicators see e.g. the reviews (and works) of Maurizio G. Paoletti, http://www.researchgate.net/profile/Maurizio_Paoletti
and Celine Pelosi
http://www.researchgate.net/profile/Celine_Pelosi



75-79
How do you define intensity?

Cattle-grazing: I have problems with this system, what you sampled is not a cattle-grazing system but two yeas old cereal fields, where before there was cattle-grazing, that is not the same thing. This should be clearly pointed out. Also in view of your statement that populations adapt rapidly to new conditions.


94-97 a map would be nice


99 C°


104, but that means lot of things, the ecological effects on soil of cow-grazing if very different than cropping cereals


103 the 3 agroecosystems need to be characterised in term of their management e.g. cows stock density, pasture management, inputs (fertilises pesticides), etc. Stock density may greatly affect soil characteristics. Pesticides and herbicides can affect earthworms. The use of heavy machinery is also known to have a negative impact on EW because of soil compaction.


119 two years


175-176 species?


209 elements with low soil mobility, example?


240-247 it would be interesting also to discuss the effect of invasive species, do they reduce the potential number of species (native disappeared), or they increased the overall number of species? (how much?) so... could we say that more species can be interpreted as a positive outcome?


272 Greek words = ?


274 Effect of pesticides/herbicides? It seems that this is a very important issue
see this recent review on the topic
Pesticides and earthworms. A review
Céline Pelosi et al. Agron. Sustain. Dev. DOI 10.1007/s13593-013-0151-z

see also this paper by Pelosi et al . that also carry on a comparative analysis of the EW assemblage in different topologies of farming systems
https://hal.archives-ouvertes.fr/hal-00886495/document
Earthworm community in conventional, organic and direct seeding with living mulch cropping systems. Agronomy for Sustainable 2009


298 ), and


303-305 it is not that simple to move from the findings of a work of this kind to the setting up of a reliable set of indicators based on EW, you may provide an attempt in this paper

Experimental design

* Cattle-grazing: I have problems with this system, what you sampled is not a cattle-grazing system but two yeas old cereal fields, where before there was cattle-grazing, that is not the same thing. This should be clearly pointed out. Also in view of your statement that populations adapt rapidly to new conditions.

* Effect of pesticides/herbicides? It seems that this is a very important issue
Pesticides and earthworms. A review
Céline Pelosi et al. Agron. Sustain. Dev. DOI 10.1007/s13593-013-0151-z

Validity of the findings

See the notes listed in Experimental design.

* It would be interesting also to discuss the effect and role of invasive species, e.g. if more species means more invasive species is that good? What's the role of invasive species in relation to the native (are the latter at risk?)

303-305 it is not that simple to move from the findings of a work of this kind to the setting up of a reliable set of indicators based on EW, you may provide an attempt in this paper

Additional comments

Due to the importance of preserving soil health, the topic is surely very important. This work then may offer some new information. However, in order to be considered for publication it requires a major revision of the present text, and some more and important details need to be supplied as well.

Reviewer 2 ·

Basic reporting

The authors present a paper where the value of earthworm communities as indicators of certain properties of soils under different agroecological systems is evaluated. The study was carried out in the Argentinian pampas, where studies are relatively scarce, and thus is worth for the general knowledge of South American earthworms.
The structure of the paper is within the standard of current scientific papers and the presentation in terms of figures and tables is ok.
The amount of sampling is overwhelming, and this guarantee that authors are presenting robust data
English needs a careful revision, as far as several spelling mistakes appear all over the text.
In section 5, the information on E. s. and A. c. (lines 284-287 and in lines 293, 296) is repeated .
There is a misspelling in line 272

Experimental design

Give more details about replicates for each treatment (are they clumped or equally distant?)
Provide agroecosystem abbreviations (NG, CG, AG) in section 2.2.
Indicate in section 2.3 the type of statistical used programs in the following cases: non-parametric ANOVA, Mann-Withnney test and diversity estimations (Chao index).
Indicate in Tables 1 and 2 the kind of variation you are presenting (SD, SE?)
Indicate in Section 2 the test used In Table 2 to compare Shannon diversity index; in this table provide also the meaning of letters.

Validity of the findings

Results are important and clearly indicate how earthworm assemblages characterize common land use systems in Argentinian pampas.

Additional comments

In enjoyed the lecture of this paper because it provides information on low studied ecosystems (as the Argentinian pampas) and how earthworms can be good indicators of soil properties for different land use systems.
My corrections are really minor ; however I suggest you to enrich the discussion by comparing your soil variable ranks of selected species (Md, Ac, Ar, Es) with more already published data.

---

## Round 0.2 · Minor Revisions

The authors have revised the manuscript and answered all the reviewers comments. But after reading the manuscript, I still found the manuscript needs improvement, especially in the description of soil properties. I have made annotations on the text.

---

## Round 0.3 · Minor Revisions

I have read and reviewed the paper again. I still feel that the English can be improved. I have made annotations again on the manuscript, but I also suggest if the authors can have a native English speaker help to edit their manuscript or hire an editing service.

---

## Round 0.4 · accepted · Accept

Thank to the authors for your submission. As the manuscript has been proofread by a few native speakers, it should be published.